# Design and Experiments of a Water Color Remote Sensing-Oriented Unmanned Surface Vehicle

**DOI:** 10.3390/s20082183

**Published:** 2020-04-12

**Authors:** Yong Li, Liqiao Tian, Wenkai Li, Jian Li, Anna Wei, Sen Li, Ruqing Tong

**Affiliations:** 1State Key Laboratory of Information Engineering in Surveying, Mapping and Remote Sensing, Wuhan University, Wuhan 430079, China; yongli0@whu.edu.cn (Y.L.); liwenkai@whu.edu.cn (W.L.); weianna@whu.edu.cn (A.W.); Lisen9368@whu.edu.cn (S.L.); 2School of Remote Sensing and Information Engineering, Wuhan University, Wuhan 430079, China; lijian@whu.edu.cn; 3School of Geodesy and Geomatics, Wuhan University, Wuhan 430079, China; tongruqing@whu.edu.cn

**Keywords:** floating optical buoy (FOBY), multi-sensors, ocean color remote sensing, skylight-blocked approach (SBA), unmanned surface vehicle, water sample collection, water spectral acquisition system

## Abstract

Integrated and intelligent in situ observations are important for the remote sensing monitoring of dynamic water environments. To meet the field investigation requirements of ocean color remote sensing, we developed a water color remote sensing-oriented unmanned surface vehicle (WC-USV), which consisted of an unmanned surface vehicle platform with ground control station, data acquisition, and transmission modules. The WC-USV was designed with functions, such as remote controlling, status monitoring, automatic obstacle avoidance, and water and meteorological parameter measurement acquisition, transmission, and processing. The key data acquisition module consisted of four parts: A floating optical buoy (FOBY) for collecting remote sensing reflectance ($R_{rs}$) via the skylight-blocked approach; a water sample autocollection system that can collect 12 1-L bottles for analysis in the laboratory; a water quality measurement system for obtaining water parameters, including Chlorophyll-a (Chl-a), turbidity, and water temperature, among others; and meteorological sensors for measuring wind speed and direction, air pressure, temperature, and humidity. Field experiments were conducted to validate the performance of the WC-USV on 23–28 March 2018 in the Honghu Lake, which is the seventh largest freshwater lake in China. The tests proved the following: (1) The WC-USV performed well in terms of autonomous navigation and obstacle avoidance; (2) the mounted FOBY-derived $R_{rs}$ showed good precision in terms of the quality assurance score (QAS), which was higher than 0.98; (3) the Chl-a and suspended matters (SPM) as ocean color parameters measured by the WC-USV were highly consistent with laboratory analysis results, with determination coefficients (R^2^) of 0.71 and 0.77, respectively; and (4) meteorological parameters could be continuously and stably measured by WC-USV. Results demonstrated the feasibility and practicability of the WC-USV for automatic in situ observations. The USV provided a new way of thinking for the future development of intelligent automation of the aquatic remote sensing ground verification system. It could be a good option to conduct field investigations for ocean color remote sensing and provide an alternative for highly polluted and/or shallow high-risk waters which large vessels have difficulty reaching.

## 1. Introduction

Coastal/inland waters have close interfaces and contribute great value to our society [1]. These waters are extremely affected by human activities due to their proximity to the human population. Accurate monitoring of water quality and environmental change is vital to various communities. Such monitoring requires access to a substantial amount of water biogeochemical information. Satellite remote sensing can provide views of aquatic environments, with advantages of synoptic coverage and repetitive observations [2]. Reliable monitoring of long-term changes or trends largely relies on the accuracy of satellite-derived data products [3].

In situ water-leaving radiance ($L_{w}$) or remote sensing reflectance (*R*_rs_) data are the premise of ocean color remote sensing modelling and the validation of satellite products, including surface Chlorophyll-a (Chl-a) concentration [4], suspended particular matter (SPM) concentration [5], and colored dissolved organic matter (CDOM). Therefore, an intelligent system that can automatically conduct a comprehensive collection of water samples and measure the water-leaving reflectance is crucial.

The floating observation method is widely used in coastal and inland water radiometric measurements, which can manually measure the radiance of the water surface and collect water samples for the laboratory analysis of marine biogeochemical parameters However, the floating observation method relies on a carrier platform. Under the existing technical background, an autonomous unmanned ship can serve as a good observation carrier. The autonomous navigation of unmanned ships combined with the floating water observation method can effectively cooperate with remote sensing monitoring. 

Unmanned surface vehicles (USVs) may be a safe and cheap solution for the multisource data of *R*_rs_ measurements [6]. Traditional methods based on survey vessels need a crew to carry many devices; moreover, the vessels’ load is heavy, making it difficult to work in shallow waters and dangerous for the crew and other staff [7]. USVs can effectively avoid these dangers and automatically complete observation tasks; in addition, USVs have been widely adopted to build a testbed for a vessel named Israeli Protector USV [8], develop equipment for maturity [6], and gather environmental data [9]. 

At present, the international ocean color community mainly promotes two methods: (1) In-water profile and (2) above-water methods [10]. However, USVs are rarely used for *R*_rs_ field measurements. In addition, the existing system for water-leaving reflectance cannot be mounted on USVs. The in-water profile method is not suitable for navigation observation, whereas the above-water method has strict requirements on observation geometry. Thus, the integration of small unmanned ships becomes a challenging task. The skylight-blocked approach (SBA) as an improved water spectrum collection method is also gradually being applied and promoted [11]. It is used for the spectral observation system of USVs because it can directly measure water-leaving radiance. 

The development of USVs can be traced back to 1898, when the famous American inventor of Serbian origin Nikola Tesla (1856–1943) invented the remote control boat named “Wireless Robot”. Before the 1980s, due to technological limitations, the development of USVs did not make a big breakthrough, mainly used for maritime targets for military exercises and artillery shooting [12]. In the 1990s, with the deepening of understanding of USVs, the potential of unmanned surface boats in anti-submarine, anti-mined maritime reconnaissance and surveillance was gradually revealed [13]. In the 21st century, with the development of communications, artificial intelligence, and other technologies, USVs have ushered in a period of rapid development [14].

To explore the possibility of USVs automatically measuring water-leaving reflectance for shallow waters, we developed a water color remote sensing-oriented unmanned surface vehicle (WC-USV) for collecting in situ water spectral and constituent data, which are fundamental for water quality remote sensing. The goal of the WC-USV is to integrate various sensors for the monitoring of shallow waters. The WC-USV is valuable to obtain automatic observations, reduce risks to researchers, free researchers from complicated in situ experiments and promote the development of ground observation systems for the development of ocean color remote sensing algorithms. The WC-USV can also serve as a ground verification system to calibrate and validate satellite remote sensing data.

The rest of this paper is organized as follows: Section 2 introduces the overall requirements and design of the WC-USV; Section 3 illustrates the major functions of the WC-USV, including measurements of water-leaving reflectance, water quality parameters, and meteorological parameters; Section 4 describes the experimental results and practical application of the WC-USV; Section 5 summarizes the study remarks and prospects for future works.

## 2. Requirements and Design of the WC-USV

The WC-USV can be used to monitor coastal/inland lakes. It can also be adopted to conduct many different types of surveys, including water-leaving reflectance acquisition, water quality monitoring, meteorological observation, and water sample collection. A traditional survey vessel needs a crew to drive the vessel and carry many devices; thus, working in shallow waters becomes difficult. To navigate in shallow water automatically, the WC-USV must have a shallow draft and a capability to avoid collision. 

These considerations led to the following requirements: (1) The hull of the WC-USV should guarantee transportation convenience and have a certain resistance to corrosion and collision. Moreover, it should have a powerful autopilot, and the draft should be less than 0.3 m. (2) The WC-USV requires a battery life of 4 hours or more, a regular speed of more than 5 km/h, and a stable autonomous navigation function. (3) The WC-USV must have modularized functions, such as collecting water-leaving reflectance, water quality, meteorological parameters, and water samples. (4) The WC-USV requires a powerful computer system which can process the collected data and ensure the normal operation of navigation and obstacle avoidance. (5) The WC-USV should have a ground system to monitor and control the WC-USV’s operating status in real time. The system should support two modes of operation: Manual remote operation and autonomous navigation. (6) A 4th generation mobile communication technology 4G communication system is needed to ensure the connections between the WC-USV and the ground system.

Consequently, the design of the WC-USV should meet the following technical and scientific demands: (1) The WC-USV can complete the measurement task in accordance with the predesigned route. After the task is completed, it can automatically return to the initial location. In addition, it can execute the task command sent by the ground system and freely switch between automatic navigation and manual control modes. (2) The WC-USV has automatic high-precision navigation and positioning functions based on the Global Positioning System GPS, electromagnetic compass, and other navigation and positioning equipment. (3) The ground system implements the control of the WC-USV and monitors the running state in real time via 4G communication. (4) The WC-USV uses microwave radar to detect and avoid obstacles. (5) When the WC-USV arrives at the planned position, the water spectral sensors are sent and withdrawn automatically, the water quality sensors start to obtain water quality parameters, and water samples are collected by the WC-USV. (6) The ground system can remotely control the WC-USV’s direction and speed; manually or automatically plan the route; display the status of the electric propeller, battery voltage, and other information; set an alarm for low battery; store the navigation log; and perform data management. 

In this study, a fully enclosed single-hull vessel, which adopted the structure of fiberglass-reinforced plastic-coated waterproof wood core material, was designed to ensure that the WC-USV would not sink when overturned. As shown in Figure 1, the 4G camera and obstacle avoidance sensor were installed in the front of the deck; the bracket was installed at the rear with the meteorological sensors, antenna, and electric winch stored; the 1.2 kW electric propeller at the stern provided the power; another bracket for fixing the spectral floating buoy and the pre-embedded connection was sealed on the tamper plate, thus ensuring that the buoy was clean and dry.

### 2.1. Architecture of the WC-USV

The WC-USV was designed as a medium waterplane single-hull vessel which was versatile in terms of mission profile and payload. Filled with polyurethane foam materials between the hull and the deck, the large and soft fender could bear approximately 100 kg of buoyancy to ensure that the vehicle did not sink. The achieved hull length, width, and moulded depth were 2.5, 1.0, and 0.43 m, respectively. The fully loaded WC-USV weighed approximately 350 kg in the air, considering the installation of the on-board equipment, which is described in the subsequent sections. The designed draft of the hull was kept low. Figure 1 shows the WC-USV’s architecture. The dimensions of the WC-USV are shown in Table 1. The schematic shown in Figure 2 displays the layout of the components within the hulls. 

The WC-USV was equipped with a GPS, three types of compasses, and a speedometer. Redundancy in the compasses was a requirement for navigation systems; these compasses located the WC-USV using a novel form of multisensor data fusion (MSDF) algorithm [6]. The proposed MSDF strategy took advantage of soft computing methodology to adapt to persistent sensor noise [15].

To minimize noise, the WC-USV propulsion system consisted of a propeller powered by a set of 24-V trolling motors. The four batteries were installed in the rear of the cabin that was placed in each hull accessed by a watertight hatch, as shown in Figure 2. These batteries were paired together to supply 24-V power through which each battery could provide a current of 30 ampere-hour (Ah). The total weight of the hull could be adjusted in accordance with the floating position of the hull. The water sampling box, host, and charger were placed in the cabin, and the rear end of the tank was close to the cabin. The wall was the lowest point of the hull; here, a bilge pump was installed to drain the water in the tank to improve the stability of the hull.

The main tasks of the WC-USV were to use the environmental monitoring sensors, as shown in Table 2, to obtain the water quality and meteorological parameters. The water quality sensor was integrated into a dedicated platform installed at the tail of the WC-USV, and the height could be adjusted in accordance with the actual situation. The probe could be replaced or cleaned, as shown in Figure 1. The drag-and-drop spectral measurement system named FOBY (floating optical buoy) was used to measure the water-leaving radiance and downwelling irradiance as the drag-and-drop method could reduce the impact of the hull shadow on the $L_{w}$ measurement. The SBA [16] could be used to eliminate the effect of skylight, measure the $L_{w}$ directly, and improve the accuracy of the R_rs_ measurement.

As shown in Figure 1, the electric propeller used was the German Torqeedo Cruise 2.0 RL with a rated speed of 1300 rpm, a weight of 16.9 kg, and a size of 240*70*120 mm. It can provide 1.12 kW of power. The WC-USV had a maximum speed of 6 km/h and a working speed of approximately 4 km/h, and could work at working speed for 5 h. In the WC-USV, four sets of measurement systems, namely, the hull position, water quality, water-leaving reflectance, and meteorological parameter systems, were integrated.

### 2.2. Ground Control System of the WC-USV

The ground control system in Figure 3a is one of the most important systems of the WC-USV. This system can plan the WC-USV route and send commands via the remote-control box and receive measurement data and hull status information from the WC-USV. A powerful computer was installed at the center of the ground control system, which was connected to a serial expansion board and network ports. The data from the measuring sensors were collected, processed, and stored using this computer. In addition, this computer could preserve data in the event of communication failure, to avoid losing data. The status check and fault diagnosis systems embedded in the ground control system could monitor the WC-USV in real time [17]. 

A communication system named Inmarsat (Fleet Broadband), which included a radio frequency (R/F) modem, was also installed in the ground control system. When the distance between the ground control system and WC-USV was less than 2 km, the WC-USV was remotely controlled by the ground control system through the Inmarsat system [18]. Microwave and 4G communication technologies were used in the transmission of digital signals and image data, respectively [19]. As a result, the WC-USV and the ground control system were connected to the Internet and could communicate in real time.

Figure 3b shows the software interface for the WC-USV installed in ground control systems. The interface was divided into four parts: (1) The top left is the main menu, (2) the left is the task planning bar, (3) the right window shows the planned sites, and (4) the bottom shows the work content for each site. The interface could also display the location of the ground control system; the status information, such as the speed heading direction and distance to target, of the WC-USV; and the measurements of spectrum, water quality parameters, and meteorological parameters in real time. Once the route was planned, it was sent to the WC-USV which executed the command immediately. The software could control the WC-USV to pause, resume, and reset the task whenever needed [20].

The ground control system could effectively control the WC-USV to operate according to the commands. In addition, the ground control system could predict alarm errors, which were handled by the emergency response system, if necessary. When power failure occurred, the ground control system could command the WC-USV to cut off all power except for the GPS and wireless transmission modules, which ensured the safety of the WC-USV.

### 2.3. Collision Avoidance System of WC-USV

WC-USVs may encounter many obstacles during navigation on water. Collision avoidance, which is achieved by using microwave radar and a 4G forward-looking camera, is necessary for WC-USVs. The automatic obstacle avoidance control module of the WC-USV can automatically avoid static or dynamic obstacles quickly. This module can process the data from the microwave radar and 4G camera to determine the location of the obstacles and adjust the direction and speed of WC-USVs to avoid them. To ensure the safe driving of the WC-USV, the obstacle avoidance system must be equipped with an improved path planning system that can respond promptly when encountering obstacles. 

WC-USVs estimate whether obstacles appear in the way of routes on the basis of the original microwave radar and 4G camera data. If WC-USVs find no obstacles, then they maintain the planned route; otherwise, they call the obstacle avoidance module and replan the route in accordance with the distance of the obstacle and the WC-USV. The steps of the function are shown in Figure 4.

## 3. Field Measurements of the WC-USV

The WC-USV carries the sensors, which use high-capacity batteries as power source. The WC-USV is small and light, making it easy to reach different types of water. The measuring systems carried by the WC-USV include the FOBY, the water quality measuring system, meteorological and hydrological automatic monitoring system, and automatic water sample collection system. For the first time, WC-USV realized the integrated automatic collection function of water spectrum, water sample, water quality, and meteorological parameters.

### 3.1. Measurements of Water Spectrum

In traditional above-water approaches, $L_{w}$ is derived from the measurements of upwelling radiance ($L_{u}$) and sky light ($L_{sky}$) by
(1)$$R_{rs}=\frac{L_{w}}{E_{s}}=\frac{L_{u}-\rho L_{sky}}{E_{s}}$$
where ρ is the Fresnel coefficient and $E_{s}$ is the downwelling irradiance just above the water surface. All relevant properties are measured from an above-surface platform and $L_{w}$ is calculated by removing surface-reflected light [21]. 

However, the above-water method is not suitable for the WC-USV because of strict requirements on observation conditions and difficulty in removing surface-reflected light. The SBA scheme, which can obtain $L_{w}$ and $R_{rs}$ directly without strict observation geometry, was adopted. 

The SBA-based FOBY is shown in Figure 5. The FOBY was integrated into a floating module that was easy to operate and avoided the influence of the WC-USV’s shadow to ensure the accuracy of the measurements of water-leaving reflectance. The upward irradiance probe and downward radiance sensor with a cover in the center of the FOBY could synchronously measure $E_{s}$ and $L_{w}$, respectively. Data processing and self-shadow correction methods are detailed in Lee’s article [16]. It was the first time the spectrum measurement system was integrated on the USV, and WC-USV realized the fully automated collection capability of the water spectrum.

### 3.2. Measurements of Water Quality and Meteorological Parameters

As illustrated in Figure 6, the sensors for measuring water quality and meteorological parameters were installed in the stern of the WC-USV and could be lowered or raised depending on the observation conditions. The water quality sensor installed at the stern could minimize the effect of water and foreign matter in the water. As shown in Table 2, the sensors could measure several important parameters, such as the water potential of hydrogen (pH), dissolved oxygen, and turbidity, which are vital for the determination of pollution level and source. Meanwhile, meteorological parameters were also measured by the WC-USV. The meteorological sensors were installed at the highest point of the hull of the WC-USV to obtain accurate meteorological data. The meteorological parameters included wind speed and direction, air pressure, and temperature and humidity. 

The initial idea was to place the water quality sensors on the bow or the bottom of the vessel, but in consideration of the impact of the water on the probe, the water quality sensors were installed at the tail of the WC-USV. In this manner, the impact of water on the probe could be reduced, and the lifespan of the sensors could be improved. The water quality sensors were submerged under water; thus, the meteorological sensors were mounted on the hull of the WC-USV. These sensors could measure the desired parameters throughout the voyage. 

### 3.3. Collection of Water Samples 

The mature water quality sampler TC-8000E was integrated into the WC-USV to collect water samples conveniently and reliably. The TC-8000E met the requirements of national environmental monitoring technical regulations. Considering the increasing needs of environmental monitoring at all levels, this sampler can set different sampling time and volume in accordance with the monitoring needs. The number of samples can be set at 1–12, and each has a capacity of 1000 mL.

The WC-USV sampling system was designed for different working modes, such as optional capacity, single-point multiple, and multipoint sampling, to meet different water sampling needs. The system could protect samples from light and heat. Such a feature was important for the laboratory accuracy of the analysis of water samples. These collected samples were crucial to verify $R_{rs}$, which is important to derive the optical and biogeochemical properties of water [4,5]. 

## 4. Tests of WC-USV

The WC-USV was designed to measure water spectral data, such as $L_{w}$ and $R_{rs}$; water quality parameters, such as Chl-a, SPM, and CDOM; and other parameters. Field experiments were conducted on 23–28 March 2018 in the shallow Honghu Lake, China, to evaluate the performance of the WC-USV. The tests were divided into two parts: Navigation and measurement. The navigation tests included a basic power test and an obstacle avoidance test. The measurement tests included water-leaving reflectance, water quality, meteorological parameters, and water sample collection. These tasks could be performed simultaneously or separately. The following test results validated the functions of the WC-USV.

### 4.1. Tests of Collision Avoidance

The tests were conducted to check the maneuverability of the WC-USV. To arrive at the designated location as quickly as possible, the working speed should be more than 4 km/h. The maneuvering tests were conducted in consideration of all the requirements mentioned above. In the experiments, the working speed ranged from 4 km/h to 6 km/h. The roll angle always influences the flexibility and stability of the vehicle; in the experiment, the maximum roll angle was 3°. These tests demonstrated that the WC-USV could work well in terms of maneuverability.

We also conducted a series of collision avoidance experiments. These tests were conducted to check the collision avoidance ability of the WC-USV (Figure 7). The location of the WC-USV was measured using the GPS. When obstacles appeared, the WC-USV could avoid them automatically and return to the planned route. In the experiment, the WC-USV was tested to avoid three big irregular obstacles. The WC-USV could avoid a single obstacle and return to the planned path quickly. When the WC-USV passed through two adjacent obstacles, it took longer because the optimal path was affected by the two objects at the same time.

During the experiments, big waves caused yaw problems to the WC-USV. After careful analysis, we found that this phenomenon happened mainly because of the following two reasons: (1) The filtering problem for obstacle detection caused by splash interference and (2) the considerable change in optimal heading angle when the obstacles got close with a wave motion, thus leading to many oscillations. Despite these problems, the experimental results demonstrated that the WC-USV could avoid obstacles well and maintain heading directions under level 5 wind (8–10.7 m/s) or below. 

### 4.2. Tests of WC-USV Measurements

The field test experiments were conducted at 30 stations in the Honghu Lake; at each station, the WC-USV worked at least 5 min. The measurements included water-leaving reflectance, water quality, and meteorological parameters, and water sample collection and could be performed simultaneously. At each station, the spectral system was measured at least five times, and each time lasted for 10 s. The water quality and meteorological parameters were measured during the time when the WC-USV arrived at and left the site, and 1000 mL of water sample was collected.

#### 4.2.1. Test of Water-Leaving Reflectance 

After the mission began, the WC-USV sailed at the highest speed. Before arriving at each station, the FOBY was released by the winch cable (Figure 8a). At this time, the speed of the WC-USV remained unchanged (Figure 8b). Upon reaching the station, the WC-USV propeller stopped working and waited for the FOBY to stabilize (Figure 8c). Then, the 50-s spectral measurement began; the measurement time could be adjusted in accordance with the actual task requirements (Figure 8d). After the measurement of the station was completed, the FOBY was withdrawn from the WC-USV, and the WC-USV proceeded to the next station at the highest speed. When all measurements were finished, the WC-USV returned to the initial location (Figure 8). 

Water-leaving reflectance in different waters of the Honghu Lake was collected via the SBA method during the tests. The $R_{rs}$ was recorded in the range of 300–1200 nm, with an interval of 3 nm. Figure 9 presents the $R_{rs}$ data and the corresponding SPM and Chl-a concentrations of the water types observed in the study area. The reflectance of remote sensing in different areas of the Honghu Lake varied significantly in the visible and near-infrared bands and had the following characteristics.

The reflectance of water in the range of 400–500 nm was relatively low, and it gradually rose with the increase in wavelength. The remote sensing reflectance showed a clear peak near 580 and 700 nm, and a valley near 680 and 760 nm. Among them, the reflection peak near 580 nm was caused by the weak absorption of algae chlorophyll and the scattering effect of cells. At approximately 680 nm, the absorption peak of chlorophyll appeared. When the density of algae was high, the reflectance of the water body appeared as a valley. At approximately 700 nm, the backscattering of suspended matter and chlorophyll increased, thereby increasing the reflectivity of water bodies. The reflection valley near 760 nm may have been caused by the strong absorption of Chl-a in the near-infrared band [22]. The reflection peak in the range of 680 nm was the most important spectral feature of algae-containing waters and could be used to determine the existence of algae and calculate the concentration of Chl-a (Figure 9a,c). 

On the basis of the laboratory analysis of the Chl-a and SPM concentrations at the corresponding stations, the following conclusions could be drawn: (1) The SPM concentration had a good relation with $R_{rs}$. As SPM concentration increased, $R_{rs}$ gradually increased [23]. $R_{rs}$ curves in Figure 9b were consistent with those of the Honghu Lake in previous studies [7]. The spectrum in Figure 9d might have been influenced by substrates, such as aquatic plants, which absorb most of the light under the water, resulting in a decrease in $R_{rs}$. Similar characteristics were also reported in waters with submerged aquatic vegetation [13,24].

Quality assurance score (QAS) [25] was calculated for the entire observation period to identify factors that resulted in uncertainties in FOBY-measured $R_{rs}$. All the QAS were higher than 0.98, which showed the high quality of $R_{rs}$ from FOBY. Overall, the test results verified that the spectral measurement system could meet the measurement requirements of water-leaving reflectance. The spectral measurement system was on board the WC-USV; the water-leaving reflectance could be measured automatically. Thus, the WC-USV equipped with FOBY could serve as a reliable solution to $R_{rs}$ measurements, especially for waters with high pollution and risk.

#### 4.2.2. Tests of Water Quality and Meteorological Measurements

Water quality parameters measured by the WC-USV were compared with those from the laboratory analysis of the simultaneously collected water samples. This comparison was performed to calibrate the performance of the WC-USV on water quality measurements. During the test, 200, 300, 400, 500, and 1000 mL concurrent water samples were collected from the five stations. 

The Chl-a and SPM concentrations were analyzed to calibrate the accuracy of the water quality sensors. The results from the WC-USV and laboratory analysis are shown in Figure 10. Through regression analysis of the Chl-a and SPM concentrations from the WC-USV and the laboratory analysis, we found that the WC-USV could measure Chl-a and SPM concentrations correctly with correlation coefficients (R^2^) of 0.72 and 0.77, respectively. This finding indicated that the WC-USV can meet the requirements of water quality measurement [26]. 

During the field test in Honghu Lake, meteorological parameters were collected in two time periods (10:30–12:20 and 14:20–16:00) with an interval of one minute. The measured data are shown in Figure 11. The trend of humidity data displays a relation with the trend of temperature. When the temperature increased, humidity decreased; when the temperature decreased, humidity increased (Figure 11a,b). The humidity reached its peak at noon within the measurement period (Figure 11b). The variation trend of wind speed was similar to that of temperature (Figure 11a,c). Air pressure decreased during the two measurement periods (Figure 11d); this result was consistent with the change in air pressure during the day. The above meteorological parameters were basically consistent with the real-time data of China Meteorological Center. These tests validated the feasibility of WC-USV’s meteorological measurement system.

## 5. Conclusions

The development of the WC-USV was the first attempt to integrate a water-leaving reflectance measurement system into a USV system, which had multiple modules. These modules included the USV platform and the ground control, spectrum measurement, water quality measurement, water sample collection, and communication systems. A collision avoidance system, which can avoid obstacles effectively, was also added to achieve the potential of the WC-USV. Comparisons between the laboratory analysis of water samples and the measurements of the WC-USV revealed that the WC-USV is a potentially precise approach to observe water-leaving reflectance, water quality, and related meteorological parameters concurrently. These measured data can be sent to the ground control station stably and efficiently through the communication system. 

In short, the WC-USV as a single system had a relatively fixed measurement system deviation. In the past in situ observations, due to different observation deviations of different observers, the deviation of the data was very accidental. The WC-USV can effectively ensure the consistency of data deviation, which is beneficial to the development of data post-processing. Furthermore, the USV can replace manual observation to ensure the safety of in situ observation staff. The WC-USV can fully automate the collection and measurement of the water spectrum, water quality, meteorological parameters, and water samples. This work provides new ideas for the intelligent field observation of water color remote sensing

However, some limitations should be addressed in future studies, and they are as follows: (1) The integration of the systems should be further considered to optimize the structure of the WC-USV. (2) Additional tests should be conducted in different water environments. (3) Extra sensors can be added to the WC-USV to improve the monitoring of waters with high pollution and risk.

## Figures and Tables

**Figure 1 sensors-20-02183-f001:**
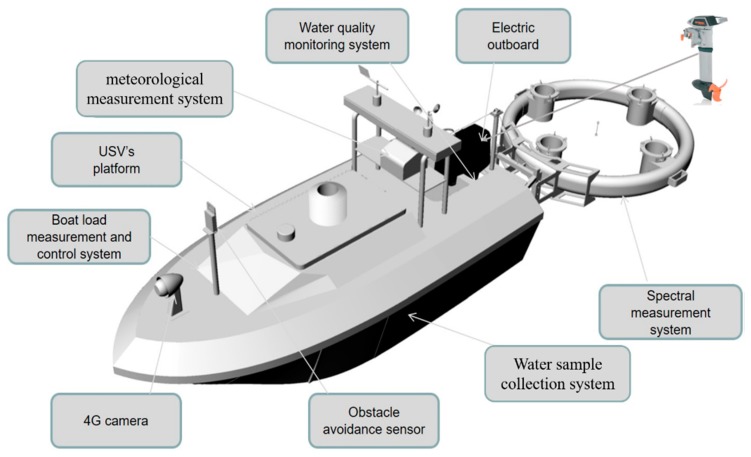
Schematic of the WC-USV.

**Figure 2 sensors-20-02183-f002:**
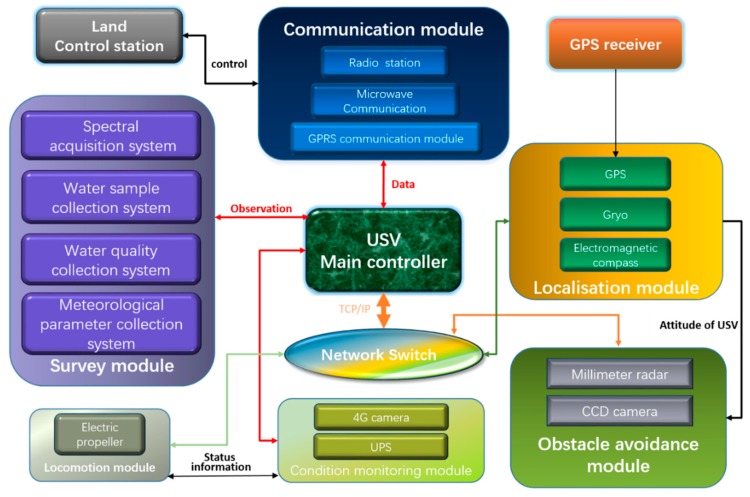
Architecture of the WC-USV.

**Figure 3 sensors-20-02183-f003:**
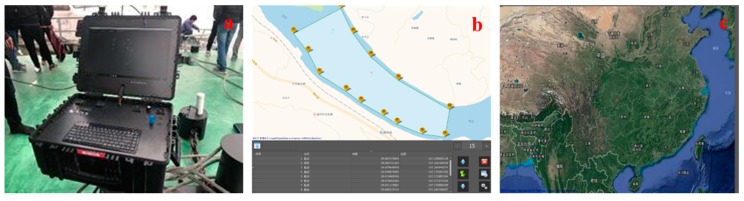
Ground control system of the WC-USV: (**a**) Control station, (**b**) route planning, and (**c**) main interface.

**Figure 4 sensors-20-02183-f004:**
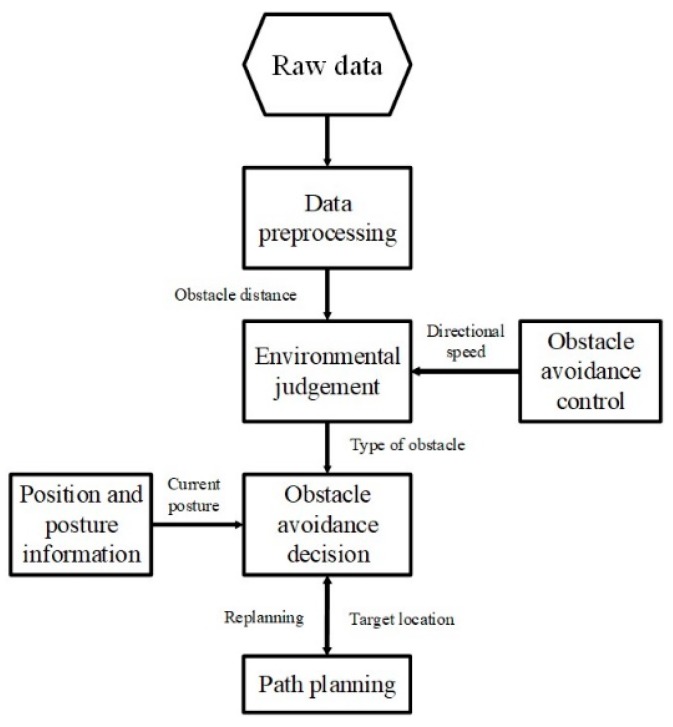
Steps of automatic obstacle avoidance.

**Figure 5 sensors-20-02183-f005:**
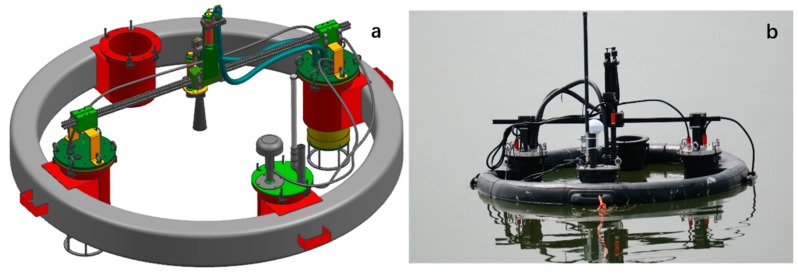
(**a**) Schematic and (**b**) field pictures of the floating optical buoy (FOBY).

**Figure 6 sensors-20-02183-f006:**
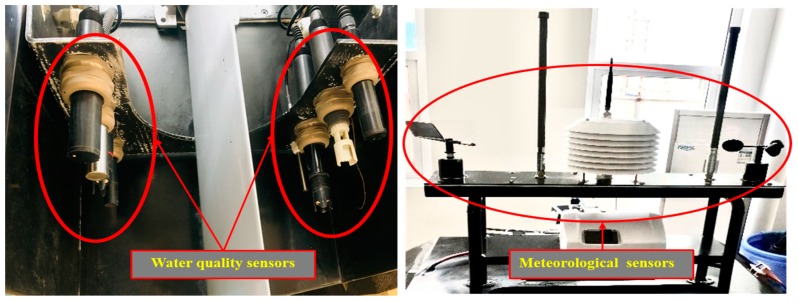
Location of the water quality and meteorological sensors.

**Figure 7 sensors-20-02183-f007:**
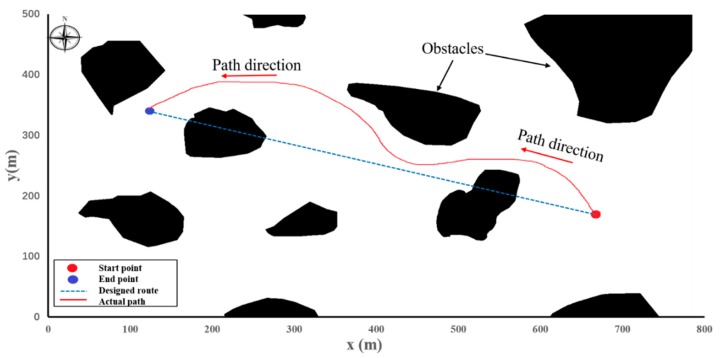
Path generated by using the WC-USV.

**Figure 8 sensors-20-02183-f008:**
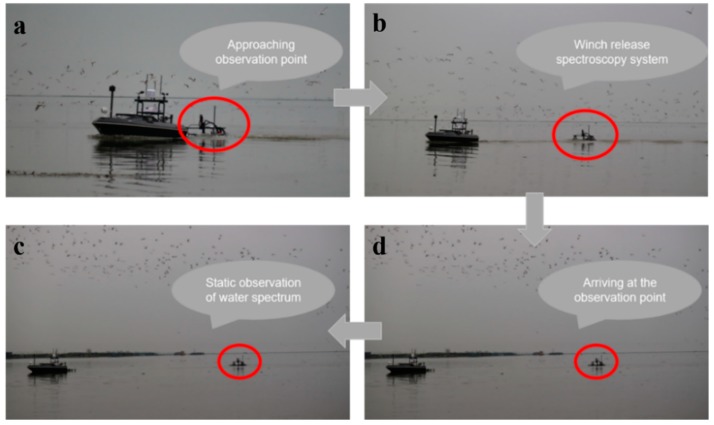
Test flow of the spectral measurement system: (**a**) The process of moving to the observation point, (**b**) emission spectrum measurement system, (**c**) Arrive at the observation station, and (**d**) spectral system starts static observation.

**Figure 9 sensors-20-02183-f009:**
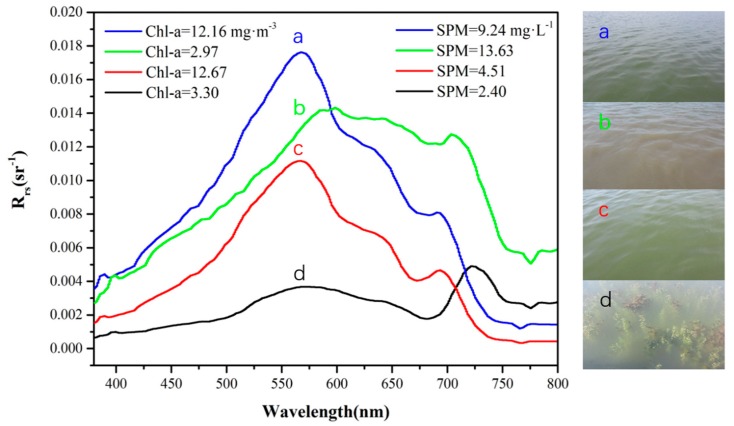
Remote sensing reflectance **(***R*_rs_)curves and corresponding Chlorophyll-a (Chl-a) and suspended matters (SPM) concentrations of different waters collected in the Honghu Lake: (**a**) Brown water with high Chl-a and SPM concentrations, (**b**) yellow water with high SPM concentrations, (**c**) green water with relatively high Chl-a concentrations, and (**d**) water with substrates.

**Figure 10 sensors-20-02183-f010:**
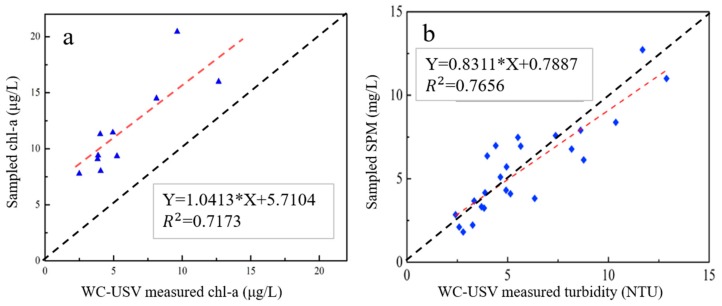
Regression analysis between the (**a**) Chl-a and (**b**) SPM concentrations measured by laboratory analysis and the WC-USV.

**Figure 11 sensors-20-02183-f011:**
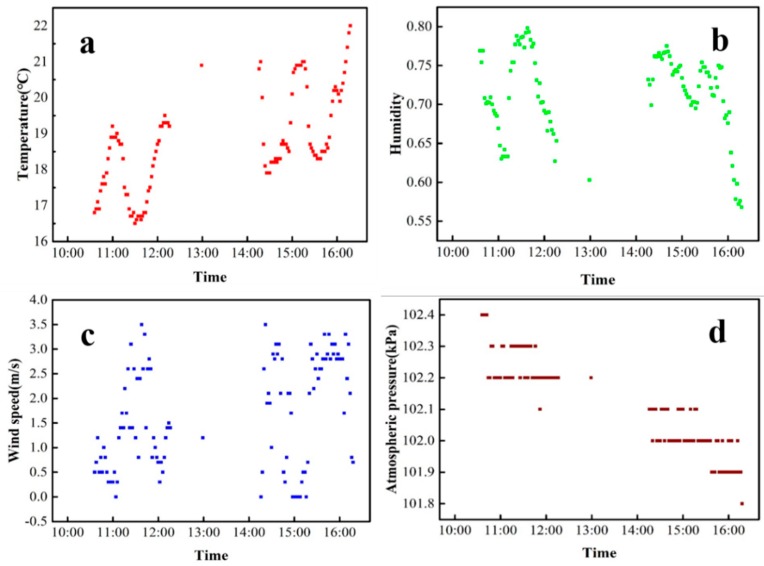
Observations on (**a**) temperature, (**b**) humidity, (**c**) wind speed, and (**d**) atmospheric pressure in the Honghu Lake on 23 March 2018.

**Table 1 sensors-20-02183-t001:** Basic parameters of the water color remote sensing-oriented unmanned surface vehicle (WC-USV).

Parameter	Value	Parameter	Value
Length	2.5 m	Total width	1.0 m
Moulded depth	0.43 m	Draft	0.25 m
Total weight	350 kg	Max speed	6 km/h

**Table 2 sensors-20-02183-t002:** Water quality and meteorological parameter monitoring sensors.

Name	Technical Index
Water quality sensors	Chl-a, turbidity, water dissolved oxygen, water conductivity, oxidation–reduction potential, water temperature, salinity, water potential of hydrogen (PH)
Meteorological sensors	Temperature, pressure, atmospheric humidity, wind speed, wind direction

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
