# Peer review of "Design and Experiments of a Water Color Remote Sensing-Oriented Unmanned Surface Vehicle"

_sensors, 2020, doi:10.3390/s20082183_

Round 1

Reviewer 1 Report

In situ measurements of remote sensing reflectance and water quality parameters are important for developing water color remote sensing algorithms as well as for validating satellite products. However, it is well known that field investigation is a very tough work. This study developed an unmanned surface vehicle platform with ground control station, data acquisition and transmission modules. This vehicle system showed reliable measurements of remote sensing reflectance, water quality and meteorological parameters, which provides great support for the development of water color remote sensing. The topic of this study is innovative, meeting the scope of the journal, and the manuscript was well written. Overall, the manuscript can be considered for publication after clarifying some minor issues as followings:

  1. Line 16, ‘developed an ocean colour remote sensing-oriented…’, suggest to change ocean color to water color throughout the manuscript. Water color includes color of both oceanic and inland waters, which is more in line with the content of this manuscript.
  2. Lines 49, 63, 70, 349, “Rrs”, italicize R
  3. Line 81, ‘shallow waters’, why is it for shallow waters?
  4. Line 101, 4 hours may be not enough? This means the USV can perform observations about 10 Km away ground?
  5. Lines 312-313, how many samples can be collected during 50 s?
  6. Line 314, it is not clear how to withdrawn the OC-USV.
  7. Line 341, QAS, avoid to use abbreviation when it appears first time in the main text.
  8. Line 345, consider to remove one ‘thus’
  9. Line 346, suggest to change perfect to reliable.
  10. Figure 10, set the same scale for x and y axes, and mention the underestimation of OC-USV. This underestimation looks like a systematic bias.

Reviewer 2 Report

It is not clear why such and expensive and massive USV was designed. It is much easier to use a normal boat and optical measurement package. What is a problem with normal boats is that you need a ramp to deploy the boat and typically two persons to do the deployment and sampling. So, one of the main advantages of USVs is their light weight and size that allows to deploy them by one person from almost any place on the shore of a lake. The USV described in this manuscript is heavier (350kg) and harder to deploy than a normal boat (=cannot be done by two people), requires special place for deployment (a large jetty or a large research vessel), has the same draft than a motorboat, and is slower than a motorboat. So, by all means it is much more cumbersome than any normal optical measurements from a boat carried out by two people. Therefore, it is hard to see what is the point in building this system.

Seems, that the study object (optical properties of optically deep and shallow lake waters) is unfamiliar for the team as they provide such a trivial results as the Figure 9 and its description. Read the deep and shallow water remote sensing literature. Then you don’t have to guess what causes the features in reflectance spectra. For example, Gitelson showed in 1992 why the Peak near 700 nm occurs and how it shifts towards longer wavelengths. MERIS (launched in 2002) had spectral bands especially selected for phytoplankton biomass detection. It takes 10-20 years to plan, build and launch a satellite. Thus, all things that the Authors guess were well known by people who decided the MERIS bands.

Row 8. The skylight blocked approach has been used for many decades and is not a novel invention. There are also many references from the last decade if you cannot get access to older publications (Kutser et al. 2013, 2016; Lee et al. 2013). I know that Swedish colleagues (Piersson and Strömbeck) used this approach in 1990s, but I could not find a reference at the moment.

Row 230. The above mentioned Kutser et al. 2013 paper is about removing sun and sky glint from the above water measurements. This study was carried out using about ten years of data where both type of measurements (above water and skylight blocked) were carried out nearly simultaneously. Thus, it is not true that the above water measurements are not suitable for the autonomous systems. Moreover, there are also automatic systems based either three Ramses or three Satlantic hyperspectral sensors that are mounted on autonomous systems keeping always right angles against sun (read several papers by Simis et al.). However, the problem with the automatically moving platforms is still the same than in the case of two sensors setup – it is hard to remove sun and sky glint.

Row 324. This is true only for strongly absorbing waters that contain either high amount of CDOM or phytoplankton (or both).

Paragraph starting from the row 324. This is a very childish story. All these facts have been known for around 40 years, Read relevant literature. There have been lake remote sensing special issues in different journals. There are basic optics books published since 1980s (Kirk, Mobley, etc.). There are several lake remote sensing review papers. Having this kind of basic description in the manuscript shows that the result came as a surprise to the Authors. Which on the other hand means that the Authors are not familiar with the field for what they try to develop instruments for.

Row 337. Read shallow water remote sensing papers about how different benthic habitats look like. There is also a recent review about this topic in the RSE.

Reviewer 3 Report

Ocean colour remote sensing-oriented unmanned surface vehicle (OC-USV) was designed and tested in this study. It is interesting for field measurement in lakes. There are some comments as follows:

(1) Discussion about the data quality of Rrs, water quality parameters should be added in the manuscript.

(2) What if the OC-USV goes into the area with fishing net or floating vegetation, please discuss them.

Specific comments:

(3) Line 260: Please give the details of the company, country, or city of TC-8000E.

(4) Line 300: Provide the “level 5 wind” with wind speed in m/s.

(5) Figure 10: Check the “Chl-a” or “chl-a” to make it consistent.

Round 2

Reviewer 2 Report

The Authors have not responded to any comments I had and therefore my recommendation is Reject, as before.

There are around 117 million lakes on Earth and only a very few of them can be studied with a system like the one proposed in this manuscript. Thus, it is pretty much useless in any lake as it requires much more effort (and money) that regular in situ optical and biological sampling carried out by people from a boat. In their response letter the Authors mention that such robotic systems may be used in dangerous conditions, like the Fukushima nuclear power plant explosion or when there are nasty bugs outside. Proper clothing is better and much cheaper against bugs. Fukushima type events are definitely one of the potential uses of the robotic system proposed, but then a completely different package of sensors is needed. The same applies to military applications where such robot may be very useful, but then it will carry guns, mines, or other military load. Thus, there are at least two potential applications where the system can be used, but those applications are not mentioned with a single word in the manuscript.

For ocean colour measurements (in lakes or shallow coastal waters) the system is far too expensive and cumbersome to use. Thus, probably nobody will use it for that purpose in real life. 

The robotic part of the manuscript is OK for me as the engineering part is well done and at least military may have interest in this system. My main concern is the ocean colour part of the manuscript. It is so obvious that the Authors do not have a glue about this topic. They do not cite relevant literature. They show obvious things that have been known for many decades. In this form the manuscript looks childish and the Authors obvious incompetence in ocean colour takes down also their achievements in robotics (that are probably not bad).

My recommendation is to write honestly that it is a navy attack and spy drone that, by the way, may have some civilian uses like water sampling in radioactive zones or ocean colour. In the current form the manuscript is absolutely unacceptable as it has ocean colour in the title, but the Authors do not have even basic understanding about the field of ocean colour. Thus, keeping the current title requires a lot of effort in reading hundreds of relevant publications from the last decades and then showing how the proposed robotic system would fit in the field. This will be much more difficult than removing the ocean colour part completely and writing about the robotics part.
